# Transcriptional Analysis of Effusion-Based Lymphoma Supports a Post-Germinal Center Origin and Specific Inflammatory Signal Background

**DOI:** 10.3390/cancers17182978

**Published:** 2025-09-12

**Authors:** Vanessa Perez-Silos, Hojung Kim, Chenguang Wang, Alejandro Zevallos-Morales, Anthony Tipton, Pierina Danos-Diaz, Ryan Wilcox, Nathanael Bailey, Nidhi Aggarwal, Savanah Dior Gisriel, Alexandria Smith-Hannah, Mina Xu, John Karl Frederiksen, Carlos Murga-Zamalloa

**Affiliations:** 1Department of Pathology, University of Illinois Chicago, Chicago, IL 60607, USA; vpsilos@uic.edu (V.P.-S.); hkim497@uic.edu (H.K.);; 2Department of Internal Medicine, University of Michigan, Ann Arbor, MI 48105, USA; 3Department of Pathology, University of Pittsburgh, Pittsburgh, PA 15261, USA; 4Department of Pathology, University of Wisconsin, Madison, WI 53705, USA; 5Department of Pathology, State University of New York Upstate Medical University, Syracuse, NY 13210, USA; smithhaa@upstate.edu; 6Department of Pathology, Yale University, New Haven, CT 06520, USA

**Keywords:** effusion-based lymphoma, HHV-8 negative, chronic inflammation lymphoma, transcriptional profile

## Abstract

Effusion-based lymphoma is a rare cancer of the B-cell type lymphocyte that builds up within body fluid spaces such as the pleural and peritoneal cavities. However, the mechanisms that underlie the development of this disease are not clear. In this study, we measured which genes were active in eight patient samples and compared these patterns with those from other well-characterized lymphomas. The gene patterns identified are most similar to those identified in large B-cell lymphomas and showed features of B cells that have matured and been repeatedly stimulated by immune responses. Taken together, these results support the idea that long-lasting inflammation in body cavities can keep stimulating B cells and further promote lymphoma to arise.

## 1. Introduction

Effusion-based lymphoma (EBL) is a rare subtype of large B-cell lymphoma that typically presents with effusions in body cavities without forming an associated solid tumor mass [1,2,3,4,5,6,7,8]. EBL usually presents in immunocompetent patients with underlying comorbidities that increase the risk of fluid overload (chronic heart failure, liver cirrhosis), with only a small subset of these patients (2–5%) being HIV-positive. Overall survival ranges from 15 months to 9 years, and the 2-year survival rate ranges from 38% to 85%. In contrast to primary effusion lymphoma (PEL), the neoplastic cells of EBL lack KSHV/HHV-8 infection, show positivity for EBV in 6–10% of cases, and express CD20 in over 95% of cases [1]. Most cases express an activated (non-germinal center) B-cell immunophenotype, with frequently detected MYD88^L265P^ mutations. In addition, the neoplastic cells feature frequent somatic hypermutations, which support a post-germinal center origin [1,9]. Similarly to aggressive B-cell lymphomas, this group of tumors has a complex genomic landscape with numerous copy number anomalies, and rearrangements of *MYC* and *BCL2* are detected in 11–21% and 9–29% of cases, respectively [9]. 

Chronic inflammatory environments arising secondary to autoimmune diseases, or specific infectious agents such as *Helicobacter pylori*, can foster the development of B-cell lymphomas through sustained antigenic stimulation and clonal expansion of B-cells [10]. In addition, the specific cytokine-related signaling observed in chronic inflammatory microenvironments promotes B-cell survival and proliferation, blunts immune surveillance, and favors the persistence of autoreactive clones [11,12,13,14]. In the appropriate tumor microenvironment, these processes can ultimately lead to B-cell lymphomagenesis [13,15]. EBL develops in effusions arising in the setting of chronic serosal irritation or fluid overload, creating a localized inflammatory niche that may support the development of clonal B-cell populations, paralleling the inflammation-associated lymphoma model [8,16,17]. 

Based on these observations, we hypothesized that EBL represents a lymphoma that follows the paradigm of inflammation-driven lymphomagenesis. Analysis of the transcriptional landscape of large B-cell lymphoma specimens allows the identification of the specific tumor cell of origin and the characterization of tumor microenvironment-related signals that operate in parallel with B-cell lymphoma progression [9,18,19,20,21,22]. In this study, we analyzed the transcriptional landscape of eight effusion specimens involved by EBL. The results suggest that EBL defines a specific expression cluster that is transcriptionally similar to those associated with LBCLs and ABC-type DLBCLs, supporting a post-germinal center cell of origin. Furthermore, the analysis detected enrichment of inflammatory signals that have been implicated in the development of B-cell neoplasms in the setting of autoimmune diseases. These findings support a model in which EBL can arise secondary to persistent inflammatory signaling. 

## 2. Materials and Methods

### 2.1. Patient Samples 

A retrospective cohort of 8 EBL cases was collected after a search from 2000 to 2024 from the pathology archives of the University of Illinois Chicago, Yale University, the University of Pittsburgh, and the University of New York Upstate. Total RNA was isolated from formalin-fixed, paraffin-embedded (FFPE) tissue samples using the AllPrep FFPE Kit (QIAGEN, Hilden, Germany; Cat. #80234), following the manufacturer’s protocol. RNA quality and concentration were assessed using the Qubit RNA Assay Kit (Life Technologies, Carlsbad, CA, USA). Gene expression was assessed using the PanCancer Immune Profiling Panel (NanoString Technologies, Seattle, WA, USA), which includes 730 immune-related genes and a total of 770 genes.

### 2.2. Gene Set Enrichment Analysis

Differential gene expression analysis was performed using the edgeR package (version 4.4.2) [23], following the methodology described by D’Angelo et al. [24]. Raw count data were normalized using the TMM method and scaled according to coding sequence length. Differential expression was assessed using the glmLRT function to compare the EBL transcripts with the LBCL cohort. Differentially expressed genes with an FDR < 0.05 were analyzed for KEGG pathway enrichment via the clusterProfiler package (version 4.14.6) [25]. Additional gene set enrichment was calculated using AUCell (v1.30.1) across the 8 samples using the SignatureDB pathway [26]. Pathway activity was summarized by computing the mean enrichment score for each gene set across samples. We observed a moderate correlation between gene set size and enrichment score (Spearman ρ = 0.28), indicating partial size-dependence. To account for potential bias introduced by gene set size, we also calculated a normalized enrichment score for each pathway by dividing its mean enrichment by the number of genes from the gene set that were present in the Nanostring panel. We finally selected the top 10th percentile for mean and normalized enrichment scores. 

### 2.3. Expression Datasets and Cancer Subtypes

Three independent publicly available datasets representing three B-cell lymphoma subtypes were included in this study. All datasets were generated using the PanCancer Immune Profiling Panel (NanoString Technologies). The following datasets were used: follicular lymphoma (FL, n = 19, GSE147125), large B-cell lymphoma (LBCL, n = 20, GSE205919), and mantle cell lymphoma (MCL, n = 14, GSE141539). Raw count data were downloaded from the Gene Expression Omnibus (GEO) database. The datasets and patient data were merged into a single expression matrix for downstream analysis.

### 2.4. Immune Set Enrichment Analysis

To assess the relative abundance of immune cell subtypes across B-cell lymphoma samples, normalized counts obtained by edgeR were transformed to log2-counts-per-million (logCPM) for downstream analysis. To estimate the pseudo-fraction values, we used a cell-type-specific gene signature to compute a z-score per sample [27]. Negative scores were set to zero, and the non-negative scores were normalized to sum to 1 for each sample. This represents the relative enrichment of each cell type per sample. 

## 3. Results

### 3.1. Patient Characteristics

A total of eight cases diagnosed as EBL from four different institutions were included. The median age at diagnosis was 86 years, the most common involved effusion site was the pleural cavity, and most of the patients (75%) had underlying conditions associated with fluid overload, such as chronic heart failure (Table 1, Appendix A).

Outcomes were available for seven of the patients. The median survival for patients who were treated only with supportive measures was 3.2 months (n = 5). One of the patients received a single dose of rituximab and underwent drainage, but died three months after diagnosis due to complications from comorbid conditions unrelated to the neoplastic process. Two patients received multiple doses of systemic chemotherapy. One was treated initially with rituximab and has remained alive for over two years. The other was HIV-positive and initially showed a complete response to R-CHOP. However, the patient subsequently expired 39 months after diagnosis due to HIV-related complications, albeit with no lymphoma recurrence after treatment (Appendix A). Representative histologic features from one case are included (Figure 1). The neoplastic cells were characterized by large atypical forms (Figure 1) that express CD20 and are negative for HHV-8. When available, most cases featured an activated (non-germinal center) immunophenotype (Figure 1). 

### 3.2. B-Cell Lymphoma-Specific Gene Expression Profiling

We profiled the transcriptional landscape of the collected cases using the Nanostring PanCancer panel, which includes 770 genes. Frequency analysis for specific cellular subtypes indicated that B-cell markers represent the highest proportion of specific immune-related gene expression across all included samples (Figure 2), supporting the adequacy of the samples in capturing transcriptional signals from the tumor cell compartment.

Using publicly annotated transcriptional data, we compared the transcriptional profiles of EBL cases with different subtypes of B-cell lymphomas. Although this panel is enriched in genes that are involved in immune responses, the differential expression analysis allowed for comparison with differentially expressed gene transcripts that are enriched in other B-cell malignancies, including mantle Cell lymphoma (MCL), follicular lymphoma (FL), and large B-cell lymphoma (LBCL). Unsupervised clustering analysis demonstrated that the EBL cases are transcriptionally closer to the LBCL group (Figure 2 and Appendix A). We also analyzed for specific lymphoma signatures using publicly available transcriptional signatures characteristic of B-cell lymphomas (SignatureDB, [20]). We searched for pathways consistently enriched across the group of samples and ranked the gene sets by their mean enrichment scores. We identified eight signature pathways enriched in the analyzed cohort (Figure 2 and Appendix A). The enriched pathways predominantly include those associated with ABC-type DLBCLs, B-cell lymphomas with plasmacytic differentiation, and activated B-/myeloma cells downstream of IRF4/MUM1 transcriptional activity [28,29]. In addition, the analysis identified pathways enriched in B-cell lymphomas with intact HLA-DR antigen presentation and in B-cell lymphomas with high-grade, poor-prognosis signatures (Figure 2). 

### 3.3. Inflammation-Related Signaling Pathway Analysis

Using reference genes for distinct cellular subsets, macrophages were the next most-enriched cellular subtypes after B-cells in the specimens (average 24%, Figure 2). T-cell lymphocytes, on average, represented 1% of the cellularity in the specimens analyzed by transcriptional data analysis. Specific transcriptional levels of chemokines have been reported in DLBCL, some of which are associated with distinct clinical outcomes [30]. Therefore, we analyzed the levels of specific chemokines in our samples in relation to the annotated transcriptional data from a cohort of DLBCL samples. Increased levels of CXCL13 are associated with the development of B-cell lymphomas in the setting of autoimmunity [12,31], but we detected no significant change in CXCL13 gene expression in EBLs compared with the control group of DLBCLs. Among the cytokines that were differentially expressed, we observed downregulation of CXCL9, whose upregulation has been associated with the development of rheumatoid arthritis-associated B-cell lymphoma [12,30,32]. CXCL9 expression is also associated with a higher frequency of cytotoxic T-cells in the tumor microenvironment of DLBCL. Although not previously noted to be upregulated in DLBCL [30], we detected upregulation of CXCL6 and its receptors, CXCR1/2, in the EBL samples. Secretion of CXCL6 in the tumor microenvironment has been linked to increased tumor-associated neutrophils [33,34], which can increase genomic instability via reactive oxygen species, and are recruited by DLBCL cells to promote proliferation [35,36]. We also observed upregulation of both IL-6 and IL-10, which have been associated with a worse prognosis in DLBCL [15]. Subsequently, we analyzed the functional significance of the differentially expressed genes with KEGG pathway enrichment in R software. The top 12 KEGG-enriched pathways are included in Figure 2 (Appendix A). This analysis identified enrichment of genes that are associated with IFN and Toll-like receptor (TLR)-mediated activation of STAT signaling pathways, and TLR-mediated NF-kB activation. 

## 4. Discussion

Similarly to prior large series, the clinical features of the EBL cases included advanced age and comorbidities associated with fluid overload, such as chronic heart failure [1]. In this cohort, only three patients received chemotherapy. One responded completely, the second died due to HIV-related complications but without lymphoma recurrence, and the third did not respond to initial therapy and succumbed due to comorbid complications three months after diagnosis. The transcriptional signals enriched in the included cases resembled those of non-germinal center type DLBCLs, consistent with other large series that feature predominantly cases with an ABC-type immunophenotype (67%–83%) [1,9]. 

Previous studies have suggested that EBL develops in the setting of a chronic inflammatory environment, leading to the expansion of clonal B-cell populations and the development of genomic alterations secondary to chronic B-cell activation [8,16,17]. Supporting this notion, a recent genomic analysis identified recurrent patterns that are characteristic of somatic hypermutation in EBL cases [9]. The current analysis is limited to a low number of cases, mostly due to the rarity of the disease. However, the gene enrichment pathway analysis highlights the differential expression of genes that correspond to cytokines associated with the development and progression of B-cell lymphomas. Loss of MHC-I expression is detected frequently in large B-cell lymphomas and is associated with worse clinical outcomes. Interestingly, the transcriptional signatures demonstrate enrichment of MHC-I-associated transcriptional programs. This observation may indicate intact expression of antigen-presenting machinery by the tumor cells, which in turn may provide sustained protection from cytotoxic immune cell surveillance. While the distinct transcriptional signatures identified support inflammation-driven biology in EBL, additional studies with a larger number and prospectively collected samples are necessary to evaluate for risk stratification or therapy selection. 

Consistent with the complex genomic landscape and prevalence of genetic aberrations identified in EBL, the transcriptional analysis detected an enrichment of high-grade signatures, including c-MYC-related signaling, which is observed in aggressive variants of large B-cell lymphomas. Despite this, most series report that EBL is generally responsive to conventional chemotherapeutic regimens or, alternatively, to effusion-directed therapies in patients for whom systemic treatment is unsuitable [9]. Although c-MYC signaling is associated with aggressive neoplasms, we recently demonstrated the prevalence of c-MYC/PLK1 signaling in nodular lymphocyte predominant B-cell lymphoma (NLPHL), an indolent B-cell lymphoma subtype [37]. Analysis of the spatial distribution and frequency of immune cell types in NLPHL identified the presence of intact T-cell responses in the immediate microenvironment, which may preserve the indolent nature of NLPHL. Similarly to NLPHL, the prevalence of cytotoxic T-cell responses in the body cavities where these cases occur (pleural, abdominal), in combination with prevalent HLA antigenic present, may facilitate an intact immune response that checks disease progression, despite the high-grade molecular signature [38]. 

## 5. Conclusions

The current findings indicate that EBL is transcriptionally close to the large B-cell lymphoma group, and is enriched in signaling pathways that are associated with ABC-type cell-of-origin. In addition, the findings demonstrate the distinct gene expression of cytokines that are associated with inflammation-related development of B-cell lymphomas, supporting the notion of the development of EBL secondary to persistent antigenic stimulation B-cell lymphoma models. 

## Figures and Tables

**Figure 1 cancers-17-02978-f001:**
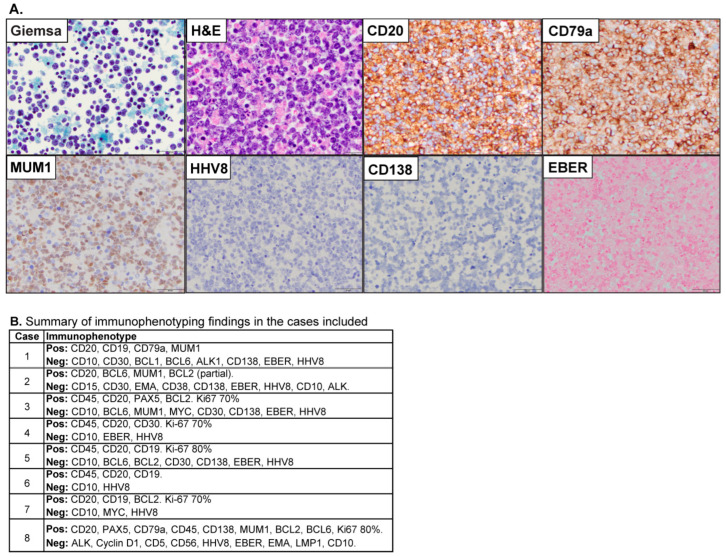
(**A**) Morphological features of a representative case of EBL are included. The pleural fluid cytology evaluation demonstrated numerous large, atypical lymphocytes with marked pleomorphism (Giemsa Stain, original magnification X40, and hematoxylin and eosin stain, original magnification X60). The neoplastic cells were positive for CD20 (original magnification X40), CD79a (original magnification X40), and MUM1 (original magnification X40), and were negative for HHV8 (original magnification X40), CD138 (original magnification X40), and EBER (original magnification X40). (**B**) Summary features for the expression of immunohistochemical markers in the cases included (Pos = positive, Neg = Negative).

**Figure 2 cancers-17-02978-f002:**
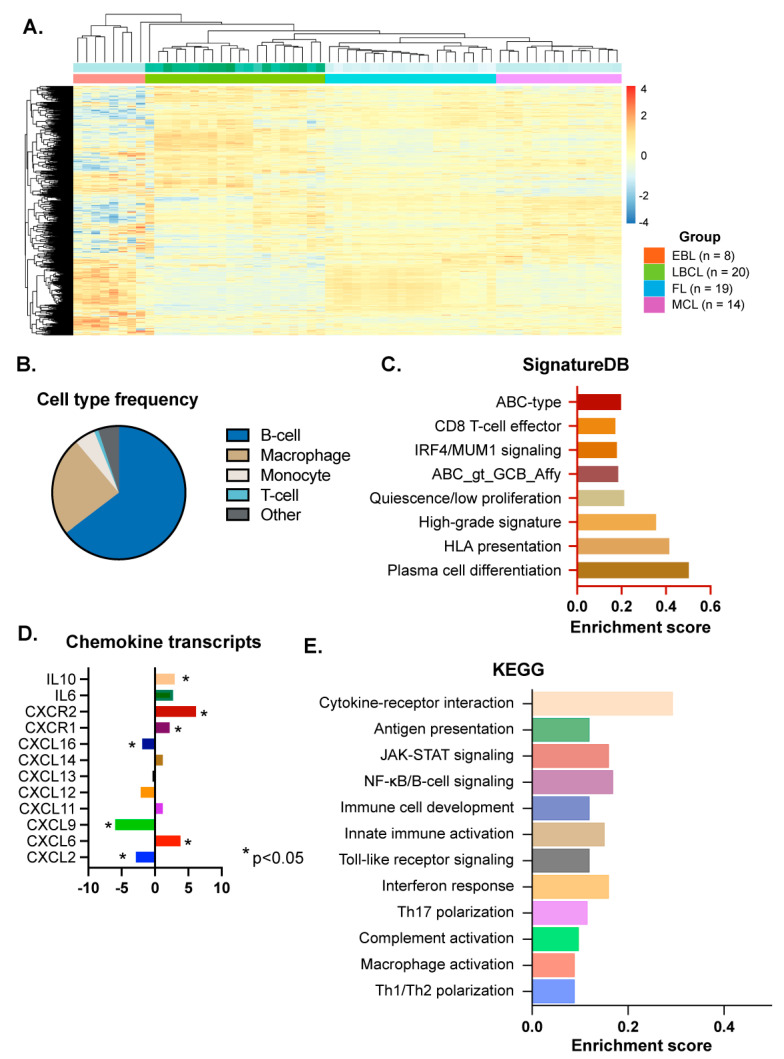
(**A**) Hierarchical clustering analysis of the 730 immune-related genes across four B-cell lymphoma subtypes. Red and blue indicate upregulation and downregulation of genes, respectively. The two bars about the heatmap represent batch and group (cancer subtype). EBL, effusion-based lymphoma; LBCL, Large B-cell lymphoma (LBCL); FL, Follicular lymphoma; MCL, mantle cell lymphoma. (**B**) Pie chart of cell type frequency in EBL cases, colored by cell type. (**C**) SignatureDB pathways. Pathway enrichment of significantly differentially expressed genes between EBL and LBCL subtypes. Enrichment score (*x*-axis) is the percentage of significant genes over the total genes in a pathway. The *y*-axis indicates pathways. (**D**) Differentially expressed chemokine transcripts between EBL and LBCL subtypes. The *x*-axis represents log2 fold change, and the *y*-axis represents chemokine transcript. *: *p* < 0.05. (**E**) KEGG pathway enrichment of significantly differentially expressed genes between EBL and LBCL subtypes. The bar plot shows selected significantly enriched KEGG pathways. The *x*-axis indicates enrichment score, and the *y*-axis indicates pathways.

**Table 1 cancers-17-02978-t001:** Summary of demographic and clinical features.

	*n* = 8
Median age (Years)	86
Male/Female	6/2
HIV	1/6 (14%)
HCV	0/6 (0%)
Risk of fluid overload	6/8 (75%)
**Effusion sites**	
Pleural total	6/8 (75%)
Pericardial total	4/8 (50%)
Pleural/Pericardial	2/8 (25%)
**Management**	
None (+/− Drainage)	4/7 (57%)
Chemotherapy	3/7 (42%)
**Responses**	
Complete remission	3/7 (50%)
Persistent or progressive	4/7 (57%)
Follow up (Median, Months)	5.3
Outcome (Deceased)	6/7 (85%)
Survival (Median, Months)	5.3

## Data Availability

The raw data supporting the conclusions of this article will be made available by the authors on request.

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
