# Peer review of "Transcriptional Analysis of Effusion-Based Lymphoma Supports a Post-Germinal Center Origin and Specific Inflammatory Signal Background"

_cancers, 2025, doi:10.3390/cancers17182978_

Round 1
Reviewer 1 Report
Comments and Suggestions for Authors
The authors used NanoString to retrospectively analyze gene expression in 8 effusion-based lymphoma (EBL) samples, and compared the results with gene expression signatures from large B-cell lymphoma, follicular lymphoma, and mantle cell lymphoma using SignatureDB and KEGG. They report that, at the transcriptomic level, EBL is more closely related to LBCL, and further analysis of immune-related pathways suggests that persistent inflammatory activation contributes to its pathogenesis. Overall, this is a concise study. While the application of a newer transcriptomic assay is appreciated, the findings largely confirm prior observations of EBL, and the degree of novelty is somewhat limited. I have a few comments and suggestions that may help to further improve the clarity and rigor of the manuscript:
- As this is a retrospective study, it would be helpful if the authors could provide the time range of sample collection.
- In the abstract, the authors refer to “Lymphoma-specific signaling pathway enrichment (SignDB),” whereas in line 175 they use “transcriptional signatures characteristic of B-cell lymphomas (SignatureDB).” Could the authors clarify whether these refer to the same Signature? There appears to be some inconsistency, as the legend of Figure 2C also uses “SignDB.”
- In Figure 2A, the red bar above the heatmap marks 9 cases. I believe the last case marked in red may actually represent LBCL. In addition, the “Group” label above the color key could potentially be removed, since the figure legend already provides the correspondence between color and group; having it on the color key may be misleading. In Figure 2C, there are two entries labeled “ABC-type.” It would be helpful if the authors could clarify whether these are distinct or duplicative.
- The authors state several times that “Unsupervised clustering analysis demonstrated that the EBL cases are transcriptionally closer to the LBCL group.” This is understandable, as both are B-cell tumors, that clustering methods may not fully distinguish them. However, it may strengthen the manuscript if the clustering results (e.g., UMAP plots or clustering scores) are provided in the supplementary material, as this would support the comparisons shown in Figure 2C–E, which are focused primarily on EBL and LBCL.
Author Response
Comments 1:The authors used NanoString to retrospectively analyze gene expression in 8 effusion-based lymphoma (EBL) samples, and compared the results with gene expression signatures from large B-cell lymphoma, follicular lymphoma, and mantle cell lymphoma using SignatureDB and KEGG. They report that, at the transcriptomic level, EBL is more closely related to LBCL, and further analysis of immune-related pathways suggests that persistent inflammatory activation contributes to its pathogenesis. Overall, this is a concise study. While the application of a newer transcriptomic assay is appreciated, the findings largely confirm prior observations of EBL, and the degree of novelty is somewhat limited. I have a few comments and suggestions that may help to further improve the clarity and rigor of the manuscript:
Response 1: We appreciate the constructive feedback from the reviewer.
Comments 2: As this is a retrospective study, it would be helpful if the authors could provide the time range of sample collection.
Response 2: Thank you for the suggestion. The pathology archives from the participating sites were reviewed to identify cases of EB-LBCL from 2000 to 2024. The methods section was modified accordingly (page 2, Line 82).
Comments 3: In the abstract, the authors refer to “Lymphoma-specific signaling pathway enrichment (SignDB),” whereas in line 175 they use “transcriptional signatures characteristic of B-cell lymphomas (SignatureDB).” Could the authors clarify whether these refer to the same Signature? There appears to be some inconsistency, as the legend of Figure 2C also uses “SignDB.”
Response 3: This was corrected across the manuscript to the corresponding designation ‘SignatureDB’ by the Lymphoma Leukemia Molecular Profiling Project.
Comments 4: In Figure 2A, the red bar above the heatmap marks 9 cases. I believe the last case marked in red may actually represent LBCL. In addition, the “Group” label above the color key could potentially be removed, since the figure legend already provides the correspondence between color and group; having it on the color key may be misleading. In Figure 2C, there are two entries labeled “ABC-type.” It would be helpful if the authors could clarify whether these are distinct or duplicative.
Response 4: Thank you for pointing out this oversight. The heatmap included in the original version of Figure 2A was a preliminary version that included a replicate case. Figure 2A has been corrected accordingly, and only 8 EBL cases are included.
Thank you for pointing out the duplicate labels in Figure 2C. Both pathways are different, and each of them has been identified to be enriched in ABC-type B-cell lymphomas. Originally, to maintain simplicity, each pathway was labeled ‘ABC-type’. To prevent confusion, the original designation of one of the pathways is now included (instead of ‘ABC-type’) in Figure 2C. The corresponding Table 2 has been modified accordingly.
Comments 5: The authors state several times that “Unsupervised clustering analysis demonstrated that the EBL cases are transcriptionally closer to the LBCL group.” This is understandable, as both are B-cell tumors, that clustering methods may not fully distinguish them. However, it may strengthen the manuscript if the clustering results (e.g., UMAP plots or clustering scores) are provided in the supplementary material, as this would support the comparisons shown in Figure 2C–E, which are focused primarily on EBL and LBCL.
Response 5: Thank you for this suggestion. We have provided a principal component analysis (PCA) of log-CPM values in the Supplementary Material (Supplementary Figure 1) to highlight the transcriptional proximity among the different samples. Given our modest cohort size and the targeted nature of the panel, we prioritized PCA over non-linear embeddings (e.g., UMAP/t-SNE), which can be sensitive to parameter choices and stochastic initialization and may yield unstable geometry in small-n settings.
Reviewer 2 Report
Comments and Suggestions for Authors
This well-written paper reports on effusion-based lymphoma (EBL), which is a rare and aggressive type of large B-cell lymphoma that presents as a body cavity effusion. The authors reviewed eight cases of EBL and compared their transcriptomes with those of other lymphomas. They identified a chronic inflammatory environment and immune response pathways associated with antigen stimulation, including Toll-like receptor and NF-κB signaling. These findings suggest that EBL is likely due to post-germinal center cells, which occur during chronic inflammation and incessant antigen exposure, characterized by an imitable gene expression pattern of inflammation-specific cytokines, contributing to the development of B-cell lymphoma. This supports the hypothesis that EBL is due to chronic antigenic stimulation. The research is well-conducted, well-documented, and methodologically sound. In addition to its scientific significance, the result will be of interest to readers and inspire research into possible treatments for this challenging lymphoma. A paradigm shift in viewing the lymphomas as secondary to chronic inflammation may bring understanding to their pathophysiology and encourage research into novel therapies.
I have no criticisms or concerns about the acceptance of the paper in its present form.
Author Response
Comment 1: This well-written paper reports on effusion-based lymphoma (EBL), which is a rare and aggressive type of large B-cell lymphoma that presents as a body cavity effusion. The authors reviewed eight cases of EBL and compared their transcriptomes with those of other lymphomas. They identified a chronic inflammatory environment and immune response pathways associated with antigen stimulation, including Toll-like receptor and NF-κB signaling. These findings suggest that EBL is likely due to post-germinal center cells, which occur during chronic inflammation and incessant antigen exposure, characterized by an imitable gene expression pattern of inflammation-specific cytokines, contributing to the development of B-cell lymphoma. This supports the hypothesis that EBL is due to chronic antigenic stimulation. The research is well-conducted, well-documented, and methodologically sound. In addition to its scientific significance, the result will be of interest to readers and inspire research into possible treatments for this challenging lymphoma. A paradigm shift in viewing the lymphomas as secondary to chronic inflammation may bring understanding to their pathophysiology and encourage research into novel therapies.
I have no criticisms or concerns about the acceptance of the paper in its present form.
Response 1: We appreciate the constructive feedback from the reviewer.
Reviewer 3 Report
Comments and Suggestions for Authors
The manuscript offers valuable insights, showing that EBL cases cluster with LBCL and are enriched in non-germinal center and inflammation-related pathways, including Toll-like receptor and NF-κB signaling. The study is timely and relevant to oncology and molecular medicine; however, several critical issues must be addressed before it is suitable for publication.
- The introduction lacks a clearly defined hypothesis. The aims should be explicitly stated in the last paragraph of the introduction section.
- The study is based on only eight EBL cases, which is a very small cohort. While understandable, given the rarity of the disease, it limits the statistical power and generalizability of the findings.
- Reliance solely on FFPE samples and a restricted Nanostring panel (770 genes) may not fully capture the transcriptomic complexity of EBL.
- Discuss more explicitly how the transcriptional findings could impact diagnosis, prognostic stratification, or therapy selection in EBL
The manuscript provides valuable insights into EBL biology but requires validation, clearer clinical integration, and refinement of presentation before being suitable for publication.
Author Response
Reviewer 3:
Comment 1: The manuscript offers valuable insights, showing that EBL cases cluster with LBCL and are enriched in non-germinal center and inflammation-related pathways, including Toll-like receptor and NF-κB signaling. The study is timely and relevant to oncology and molecular medicine; however, several critical issues must be addressed before it is suitable for publication.
Response 1: We appreciate the constructive feedback from the reviewer.
Comment 2: The introduction lacks a clearly defined hypothesis. The aims should be explicitly stated in the last paragraph of the introduction section.
Response 2: Thank you for this suggestion. A clearly defined hypothesis has been included in the introduction section (page 2, line 68).
Comment 3:The study is based on only eight EBL cases, which is a very small cohort. While understandable, given the rarity of the disease, it limits the statistical power and generalizability of the findings.
Response 3: As the reviewer indicated, this is a limitation of the study. To address this comment, the discussion section has been modified (page 7, line 225).
Comment 4: Reliance solely on FFPE samples and a restricted Nanostring panel (770 genes) may not fully capture the transcriptomic complexity of EBL.
Response 4: As the reviewer has indicated, transcriptome-wide RNA-seq is the modality of choice for studies aimed at discovering the transcriptomic complexity of specific neoplasms. The current study was designed to test the hypothesis that EBLs arise in an inflammation-driven context. In these instances, targeted expression profiling is appropriate and offers practical advantages, as it achieves greater depth on specific pathways and detects low-abundance transcripts consistently with fewer reads. Furthermore, this approach is advantageous for archival FFPE tissue, as it reduces RNA input and directly counts transcripts without reverse transcription or PCR, making them robust for degraded or FFPE-derived RNA.
Comment 5: Discuss more explicitly how the transcriptional findings could impact diagnosis, prognostic stratification, or therapy selection in EBL.
Response 5: Thanks for this suggestion. To address this comment, a new paragraph in the discussion section was added (page 7, line 233).
The manuscript provides valuable insights into EBL biology but requires validation, clearer clinical integration, and refinement of presentation before being suitable for publication.
Round 2
Reviewer 3 Report
Comments and Suggestions for Authors
Recommended for the publication.